# Trait Anxiety Leads to “Better” Performance? A Study on Acute Stress and Uncertain Decision-Making

**DOI:** 10.3390/bs14121186

**Published:** 2024-12-12

**Authors:** Yuxuan Yang, Bingxin Yan, Kewei Sun, Di Wu, Cancan Wang, Wei Xiao

**Affiliations:** Department of Military Medical Psychology, Air Force Medical University, Xi’an 710032, China; yangyuxuan@fmmu.edu.cn (Y.Y.); yanbxin@fmmu.edu.cn (B.Y.); sunkewei@fmmu.edu.cn (K.S.); wudi0426@fmmu.edu.cn (D.W.); wangcan@fmmu.edu.cn (C.W.)

**Keywords:** acute stress, decision-making, ambiguity, risk, anxiety

## Abstract

In uncertain situations, individuals seek to maximize rewards while managing risks. Yet, the effects of acute stress and anxiety on decision-making in ambiguous and risky contexts are unclear. This study aims to contribute to the exploration of how acute stress influences sensitivity to immediate vs. delayed rewards, risk management strategies, and the role of anxiety in these processes. This study used the laboratory acute stress induction paradigm to analyze the direction of influence of acute stress on ambiguity decision-making and risky decision-making in males and then used moderating effect analysis to study the impact of anxiety on this process. The results show that a combination of the Socially Evaluated Cold Pressor Test and the Sing-a-Song Stress Test can successfully induce acute stress, which reduces both the proportion of the options selected that represent long-term rewards and risk-adjustment ability. Additionally, trait anxiety had a moderating effect on the influence of stress on ambiguity decision-making. Acute stress reduces focus on long-term rewards while increasing focus on short-term rewards, leading to impulsivity and impaired risk-adjustment. Additionally, to some extent, high trait anxiety scores predict better performance in making decisions under ambiguity during stress.

## 1. Introduction

In our daily lives, we frequently confront a multitude of uncertain decisions, ranging from selecting which answer to choose during an exam to making significant life choices such as determining which job to pursue or which project to invest in. Sometimes, a choice may be good in the present, but it may not necessarily be beneficial in the long run. These decisions often need to be made under stressful conditions, and individuals need to make choices that maximize their returns, which can exacerbate the complexity of the decision-making process itself, and sometimes, the decision-making process itself can constitute a stressor [1]. One common phenomenon observed is “choking under pressure,” where individuals underperform in high-stakes situations compared to their usual performance levels without pressure [2].

Within the context of the non-specificity of stress responses, there also exists specificity [3]. This means that when individuals confront stressors, their forms of stress response can vary significantly, influenced by variations in the stressors themselves or individual differences. This variability leads us to explore several key questions: What kinds of performance differences arise among individuals when facing uncertain decisions under stress? What is the relationship between these performance differences and the characteristics of the individuals themselves? Can we predict an individual’s performance under stress based on some pre-task measurements?

To set the stage for our study, we first elucidate the concept of acute stress and the classification of decision-making under uncertainty. Subsequently, we review studies examining the role of acute stress in two types of decision-making scenarios, and we introduce the role of anxiety, one of the most significant emotions influencing the relationship between stress and cognition. Finally, we summarize the current evidence and present the objectives and hypotheses of our study.

### 1.1. Stress

To understand these questions, we must first define stress. Stress is a series of physiological and psychological responses generated by the organism to maintain homeostasis when confronted with a threat [4]. Depending on the duration of the stressor, it can be classified into acute stress or chronic stress. The physiological changes in individuals are mainly manifested in the sympathetic adrenomedullary system (SAS) and the hypothalamus–pituitary–adrenal axis (HPA-axis) [5]. The SAS, serving as a fast-responding pathway, is activated through in-creases in heart rate, skin conductance, blood pressure, and salivary amylase levels [6]. The HPA-axis, as a slow-responding pathway, is primarily reflected in elevated cortisol levels [7]. The magnitude and direction of the effects produced by stress and related hormones are not uniquely determined [8,9,10].

### 1.2. Decision-Making Under Uncertainty: Ambiguity vs. Risk

As an important and advanced cognitive activity, decision-making is a topic of great concern in both psychology and economics. Based on whether the outcome of a decision is certain and whether the task structure is clear, decisions can be classified into uncertain decision-making or certain decision-making. Uncertain decision-making can be further divided into risky decision-making and ambiguity decision-making, according to the concepts of “measurable uncertainty” and “immeasurable uncertainty” proposed by Knight [11]. Risky decision-making refers to decisions made by decision-makers when the future situation is not fully certain, but the consequences of various decisions and their probabilities are known [12]. Ambiguity decision-making, on the other hand, refers to decisions where decision-makers cannot predict the possible outcomes and their probabilities and can only rely on subjective judgments for prediction [13].

### 1.3. Stress and Decision-Making Under Uncertainty

Our research focuses specifically on the impact of acute stress on decision-making under uncertainty. Existing research in this field primarily examines the decision-making outcomes influenced by stress and the specific mechanisms that affect the cognitive components of decision-making, focusing on aspects such as learning from feedback, attention to rewards and risks, and other relevant factors [14,15].

In the study of ambiguity decision making, Shen and colleagues summarized the impact of stress on the cognitive components of ambiguity decision-making into three aspects: learning feedback, reward sensitivity, and risk preference [14]. Stress may hinder effective learning from decision feedback, ultimately reducing decision quality and weakening cognitive control [16]; however, it may also enhance individuals’ attention levels, accelerate learning feedback, and stimulate arousal, leading to im-proved decision performance [17]. Stress can also increase decision-makers’ sensitivity to rewards and decrease their aversion to losses [18]; conversely, some studies have found that stress reduces reward-seeking behavior and alleviates aversion to losses [19]. In the study of risky decision-making, acute stress has been shown to induce preferences for or aversions to risky options in decision-making tasks, exerting positive or negative effects on the perception of and attention to information, memory, executive functions, and feedback learning in the process of risky decision-making [20,21]. These effects are mainly manifested in the focus on potential risks and rewards, risk-taking during the decision-making process, and the feedback processing of outcomes [22,23,24]. The phenomenon of stress promoting risk preference or risk aversion yields inconsistent results [15,25].

Among these, an important component that deserves special attention and is common to both ambiguity and risky decision-making is individuals’ focus on reward under stress. First, individuals need to make choices that maximize returns in uncertain situations. From a temporal perspective, they must navigate the conflict between smaller but immediate rewards and larger but delayed ones [26]. From an economic standpoint, individuals frequently confront scenarios where they must weigh gains against losses. Research suggests that stress, through mechanisms involving cortisol release and information processing, can enhance individuals’ adaptability in uncertain situations, increasing their focus on and learning from rewards [27]. This is aligned with the “STARS” (Stress Triggers Additional Rewards) theory, which proposes that stress facilitates learning from positive feedback (e.g., gains) and hinders learning from negative feedback (e.g., losses) [27]. Additionally, some studies have found that stress heightens sensitivity to immediate rewards, prompting individuals to prefer smaller, immediate rewards, which can lead to poorer decision-making performance in tasks [28]. Considering these findings, it becomes evident that since positive feedback can be categorized into delayed and immediate gains, and negative feedback can be divided into small gains and losses, conclusions drawn from different studies vary [23,26]. Therefore, in this study, to distinguish individuals’ sensitivity to losses versus gains and to short-term (immediate) versus long-term (delayed) rewards, we employ a combined analysis of three paradigms to elucidate how stress influences decision-making under uncertainty.

Furthermore, gender is a key variable that also deserves special emphasis as one of the significant factors influencing the difference between acute stress and decision-making. Existing studies have confirmed that estrogens are closely related to HPA axis activity and stress-related manifestations [29,30,31] and that sex hormones from the hypothalamic–pituitary–gonadal (HPG) axis interact with glucocorticoids from the HPA axis, exerting various effects on behavior [32]. When examining ambiguity decision-making using the Iowa Gambling Task (IGT), it was found that males exhibited poorer decision-making performance, while females performed better under stress [33]. Another study also found that males were more likely to choose high-uncertainty options compared to females [34]. Significant gender differences were also observed in risk decision-making tasks. In the balloon analogue risk task (BART), females became more conservative and risk-averse under stress, while males became more risk-seeking [35]. Numerous compelling pieces of evidence suggest that gender is a critical variable that needs to be taken into account in studies on the impact of acute stress on reward processing and decision-making. Since gender effects are unclear and, moreover, difficult to control in the case of females, we focus on male participants in the sequel.

These studies have provided much evidence and understanding of the impact of acute stress on uncertain decision-making, but the conclusions vary due to differences in stress induction methods, cognitive tasks, participant groups, and other factors. Most important of all, they did not explore the impact of stress on decision-making under uncertainty by integrating ambiguity decision-making with risky decision-making.

### 1.4. Anxiety, Stress, and Cognition

Many studies have explored the impact of stress on decision-making, incorporating individual differences such as personality and behavioral styles [36,37]. Anxiety, as a significant emotion, greatly influences the relationship between stress and cognition. The “Model of Interaction between Emotion and Metacognition” suggests that decision-making is constrained by personality while also being influenced by contextual factors such as emotions [38]. Anxiety can be divided into trait anxiety and state anxiety [39,40]. Trait anxiety is a relatively stable personality trait with individual differences in response tendencies. In contrast, state anxiety is a transient emotional experience accompanied by physiological arousal and subjective feelings which varies significantly with changes in the environment. Research has found that trait anxiety is associated with increased amygdala activation and heightened fear expression during fear acquisition [41,42], which can be regulated through conscious cognitive strategies [43,44]. Anxiety is positively correlated with psychological stress responses [45], and the introduction of anxious emotions can lead to risk-averse decision-making [15]. Although there are some studies that analyze anxiety and cognitive function together, there is still limited and controversial research on how these two types of anxiety affect uncertain decision-making under stress.

### 1.5. Purpose and Hypotheses

Contributing to the discussion, this study aims to explore the impact of acute stress on uncertain decision-making, incorporating anxiety as a factor for joint analysis. By doing so, we hope to gain a deeper understanding of the influence of individual differences on cognitive characteristics under acute stress. This study may provide a certain degree of reference for predicting individuals’ performance in different situations and tasks in practice and for developing strategies to mitigate the negative effects of stress on decision-making.

We hypothesize that acute stress impacts performance in ambiguity decision-making by reducing an individual’s sensitivity to rewards and ability to learn from feedback, while simultaneously promoting an individual’s risk preference and propensity for risk-taking. Consequently, this leads to a decline in the individual’s capacity for learning and accurate judgment under uncertain decision-making conditions, resulting in short-sightedness and risky behaviors. Furthermore, we anticipate that state anxiety and trait anxiety play a moderating role in decision-making under stress and control conditions. Individuals with higher anxiety scores may exhibit poorer decision-making performance and greater impulsivity under stress.

## 2. Methods and Procedures

### 2.1. Participants

This study recruited 64 male university students as participants (M_age_ = 20.25, SD_age_ = 1.65). To avoid hormonal fluctuations influenced by menstrual cycles in females, only males were selected. To mitigate the impact of diurnal cortisol rhythms, all experiments under stress were conducted between 14:00 and 18:00. Among them, 4 were excluded for incomplete participation and 3 for contaminated or lost saliva samples, resulting in 57 valid samples for analysis. Sample size determination was based on previous studies with inter-group sample sizes ranging from 40 to 113 participants [24,26,28,46]. All participants had normal or corrected-to-normal vision, no history of neurological or psychiatric disorders, no recent use of psychoactive drugs, and no prior participation in similar experiments. Within one week prior to the experiment, all participants completed a series of questionnaires, including the Chinese Big Five Personality Inventory-Brief Version (CBF-PI-B) [47]. Subsequently, 28 participants (M_age_ = 20.32, SD_age_ = 1.63) were randomly assigned to the stress group, and 29 (M_age_ = 20.17, SD_age_ = 1.69) to the control group.

### 2.2. Measures

#### 2.2.1. Acute Stress-Evoking

Given the unsatisfactory response to a single SECPT in our preliminary tests and considering individual differences among participants, the study combined the Socially Evaluation Cold Pressure Test (SECPT) [48] with the Sing-a-Song Stress Test (SSST) [49] to induce acute stress, both paradigms having been widely used in previous research. We used E-prime 3.0 to compile a standardized procedure so that the stress conditions and control conditions correspond equally.

The stress group was informed that their performance during the tasks would be video-recorded for subsequent facial analysis, and their payment would depend on task performance (though this was not actually implemented). Participants were instructed to follow a series of prompts on the screen, each followed by a countdown timer (details in the Appendix A). They were required to immerse their left hand in cold water at 0–4 °C and read a series of neutral sentences, then sing a song aloud after the countdown, subsequently removing their hand from the water. The hand immersion lasted for 3 min, but participants could remove their hand at their discretion. The control group immersed their hand in water at 30–35 °C and read neutral sentences, but to avoid any influence of vocalization on heart rate, they were asked to read specified neutral paragraphs. Additionally, the stress group was video-recorded throughout the task and observed by a neutral experimenter, while the control group had no observation. The rest of the procedure was identical for both groups.

#### 2.2.2. Related Scales

The State–Trait Anxiety Inventory (STAI) consists of two subscales with a total of 40 items: the State Anxiety Inventory (S-AI) and the Trait Anxiety Inventory (T-AI), each containing 20 items. It employs a 4-point rating scale (S-AI: 1 = Not at all, 4 = Very much so; T-AI: 1 = Almost never, 4 = Almost always) to reflect the degree of state or trait anxiety [50]. The Cronbach’s α coefficient for homogeneity reliability is 0.906 for S-AI and 0.882 for T-AI. The test–retest reliability values are 0.88 for S-AI and 0.9 for T-AI, respectively. The correlation coefficients between S-AI and T-AI in the initial and retest measurements are 0.84 and 0.77, respectively. The reliability and validity are all satisfactory. In the subsequent analysis, we focused on the sum scores of the two subscales.

### 2.3. Uncertain Decision-Making Tasks

#### 2.3.1. Farming on Mars Task

The Farming on Mars task is an empirically-based measure of ambiguity decision-making [51]. The task was coded using the Visual Studio platform with the C# programming language. Participants were instructed to test either oxygen production system A or B in each trial, with small bar graphs on the right indicating current and cumulative oxygen extraction. Their task was to repeatedly choose which system to operate to maximize total oxygen production over the experiment (Figure 1a). System A (increasing option) offered less immediate oxygen but increased production for the next two choices, leading to greater rewards, while System B (decreasing option) provided more oxygen each time but decreased production for the subsequent two choices. In each trial, B offered a higher reward initially, but as the number of A choices increased in the last 10 trials, the rewards for both options increased. The reward score formula was: A = 400 + 1000 × h/10, B = 900 + 1000 × h/10, where h was the number of A choices in the past 10 trials (initially set at 5) (Figure 1b). With a total of 250 trials, each set of 50 trials constituted a block, and a total of 5 blocks were analyzed. Participants were not informed of the rewards or the underlying structure. There was no fixed choice time per trial, and the inter-trial interval was 750 ms. The dependent variable was the proportion of selections of the increasing option, i.e., the ratio of the number of times option A was chosen to the total number of selections.

#### 2.3.2. Cambridge Gambling Task

The Cambridge Gambling Task (CGT) measures risky decision-making [52]. The task was completed through programming using the C++ language. Participants guessed whether a gold coin was hidden behind a red or blue square on the screen and placed a bet, starting with a principal of ¥100. Nine red-to-blue square ratios (9:1, 8:2, 7:3, 6:4, 5:5, 4:6, 3:7, 2:8, and 1:9) and five bet-to-principal ratios (5%, 25%, 50%, 75%, and 95%) were used, with bet increases or decreases randomly presented across 36 trials (Figure 2).

Dependent variables included quality of decisions, decision time, rational adventure index, risk-taking index, impulsivity index, and risk-adjustment [53]. Quality of decisions was measured by the number of times participants chose the more frequent color across trials (excluding 5:5). Decision time was the duration from square appearance to color selection. The rational adventure index was the mean bet percentage for choosing the majority color (excluding 5:5). The risk-taking index was the mean bet percentage per trial. The impulsivity index was the difference between mean bets in descending and ascending order. Risk-adjustment reflects adjustments in betting proportions with changes in reward–punishment probabilities (color ratios), calculated as [2 × (bet % at 9:1) + (bet % at 8:2) − (bet % at 7:3) – 2 × (bet % at 6:4)]/mean bet %.

#### 2.3.3. Iowa Gambling Task

The Iowa Gambling Task (IGT) assesses ambiguous and risk-based decision-making [54]. The task was completed through programming using the C++ language. Four decks were presented, each offering a fixed gain (Decks 1 and 2: ¥100, Decks 3 and 4: ¥50) but with varying loss probabilities (Deck 1: 50% chance of losing ¥150–350, Deck 2: 10% chance of losing ¥1250, Deck 3: 50% chance of losing ¥20/50/75, Deck 4: 10% chance of losing ¥250) (Figure 3). Participants aimed to maximize gains. The dependent variable was the net score, calculated as [(Deck 3 + Deck 4)—(Deck 1 + Deck 2)]. There were 100 trials, with the first 40 assessing ambiguity decision-making and the last 60 assessing risk-based decision-making.

### 2.4. Acute Stress Indicators

#### 2.4.1. Psychological Assessment Indicators

Participants were asked to report their subjective stress scores at baseline, after the induction manipulation, and after the decision-making task. The Visual Analogue Scale (VAS) was used, which involved participants adjusting a 100 mm sliding scale to indicate their current level of stress. The scale ranged from “0—No Stress at All” at one end to “10—Extremely High Stress” at the other.

#### 2.4.2. Physiological Assessment Indicators

Salivary cortisol, a glucocorticoid secreted by the zona fasciculata of the adrenal cortex, is a crucial indicator for assessing HPA-axis activity. Using Sarstedt’s Salivette collection devices, participants were instructed to place a swab under their tongue for 60 s to stimulate saliva secretion at baseline and after the decision-making task. Saliva samples were then stored at −20 °C. Later, cortisol concentrations in the saliva were measured using Enzyme-Linked Immunosorbent Assay (ELISA) in the laboratory.

Heart rate, an indicator of sympathoadrenal activation, was measured using the Psychorus offline wristwatch from Beijing Huixin. Conductive gel was applied to the back of the participant’s left wrist, and the watch was worn with its metal sensor in close contact with the skin. Heart rate was assessed via photoplethysmography (PPG) at a 20 Hz sampling rate during baseline, induced manipulation period, and decision-making task period.

Blood pressure measurement, which includes systolic pressure and diastolic pressure, serves as an indicator of sympathoadrenal medullary system activation. The Omron J760 electronic sphygmomanometer was used for measurement, with the position standardized to the right arm of the participant, ensuring that the center of the upper arm cuff was at the same level as the heart. Measurements were taken at baseline, induced manipulation, and after the decision-making task.

### 2.5. Procedure

The overall experimental procedure was divided into two parts. Prior to the experiment, participants were required to complete a series of questionnaires, including the Chinese Big Five Personality Inventory-Brief Version (CBF-PI-B) [47], among others, and were then randomly assigned to groups. During the experimental phase, participants were instructed to avoid strenuous physical activity, eating, and consuming beverages other than water for one hour prior to arriving at the laboratory. Upon arrival, they rinsed their mouths, washed their hands, and rested for 5 min during which time they completed the State-Trait Anxiety Inventory (STAI). Following the rest period, baseline heart rate was measured for 3 min, followed by the collection of blood pressure, salivary cortisol, and subjective stress scores. Subsequently, participants underwent either a stress-inducing task or a control procedure, after which the indicators were repeated. They then sequentially completed decision-making tasks in a Latin square order. Upon completion of all tasks, indicators were collected again, along with a second salivary cortisol sample. Interviews were also conducted to explore participants’ experiences and observed patterns during the tasks (Figure 4). This study was approved by the Ethics Committee of Xijing Hospital (2024–06–03; KY20242146-F-1), and was performed following the ethical standards as laid down in the 1964 Declaration of Helsinki and its later amendments or comparable ethical standards, and all participants voluntarily took part in the experiment after signing an informed consent form.

### 2.6. Data Analysis

The data were analyzed using SPSS 26 and SPSSAU 24.0 software. For the acute stress indicators, a two-factor repeated measures ANOVA with condition and time as factors was conducted for salivary cortisol, heart rate, systolic blood pressure, diastolic blood pressure, and stress scores. For the decision-making tasks, repeated measures ANOVA and independent-samples *t*-tests were used to analyze the three types of tasks. Subsequently, correlation analysis and analysis of moderating effects were conducted to explore the role of anxiety in decision-making under stress.

## 3. Results

### 3.1. Balance Analysis

To determine whether there were any intergroup differences in factors such as personality and anxiety between the participants assigned to the stress and control groups, independent-samples t-tests were conducted sequentially. The results showed no significant differences in personality and anxiety (*p* > 0.1), indicating that the two groups of participants came from the same population.

### 3.2. Acute Stress Induction Analysis

#### 3.2.1. Salivary Cortisol

A two-factor repeated measures ANOVA with 2 (condition) × 2 (time) was conducted on salivary cortisol levels while incorporating the total STAI score as a covariate (Figure 5a). The results showed a significant main effect of condition (*F*(1,54) = 8.438, *p* = 0.005, *ŋ_p_*^2^ = 0.135), with higher cortisol levels in the stress group compared to the control group. The main effect of time was marginally significant (*F*(1,54) = 3.195, *p* = 0.079, *ŋ_p_*^2^ = 0.056), with baseline levels significantly lower than after the decision-making task.

There was a significant interaction effect (*F*(1,54) = 9.295, *p* = 0.004, *ŋ_p_*^2^ = 0.147). Further simple effects analysis revealed that there was marginal significant difference in cortisol levels between the groups at baseline (*F*(1,54) = 3.304, *p* = 0.075, *ŋ_p_*^2^ = 0.058), but there was a significant difference between the two groups after the decision-making task (*F*(1,54) = 9.946, *p* = 0.003, *ŋ_p_*^2^ = 0.156). Additionally, within the stress group, there was a significant difference in cortisol levels before and after the decision-making task (*F*(1,54) = 11.55, *p* = 0.001, *ŋ_p_*^2^ = 0.176), whereas no significant difference was observed in the control group.

#### 3.2.2. Heart Rate

A two-factor repeated measures ANOVA with a 2 (condition) × 3 (time) was conducted on heart rate while incorporating the total STAI score as a covariate (Figure 5b). The results revealed a significant main effect of time (*F*(2,53) = 24.243, *p* < 0.001, *ŋ_p_*^2^ = 0.478), with baseline significantly higher than the decision-making task period (*p* < 0.001) and the induced manipulation period significantly higher than the decision-making task period (*p* < 0.001). However, the main effect of condition was not significant (*F*(1,54) = 0.799, *p* = 0.375, *ŋ_p_*^2^ = 0.015).

The condition × time interaction was not significant (*F*(2,53) = 1.955, *p* = 0.152, *ŋ_p_*^2^ = 0.069). In order to further elucidate the relationships at different levels, we still employed simple effect analysis for investigation. For the stress group, the simple effect was significant (*F*(2,53) = 12.372, *p* < 0.001, *ŋ_p_*^2^ = 0.318), with baseline lower than the induced manipulation period (*p* = 0.067) and the induced manipulation period significantly higher than the decision-making task period (*p* < 0.001). Additionally, baseline was significantly higher than the decision-making task period (*p* = 0.006). We conclude that there is a certain impact on the induction of heart rate, but the performance is suboptimal.

#### 3.2.3. Blood Pressure

A 2 × 3 two-factor repeated measures ANOVA was conducted on systolic blood pressure (SBP) while incorporating the total STAI score as a covariate (Figure 5c). The results indicated that the main effect of time was not significant (*F*(2,53) = 1.161, *p* = 0.321, *ŋ_p_*^2^ = 0.042), while the main effect of condition was significant (*F*(1,54) = 5.926, *p* = 0.018, *ŋ_p_*^2^ = 0.099), with the stress group showing overall significantly higher SBP than the control group. The interaction between condition and time was significant (*F*(2,53) = 3.796, *p* = 0.029, *ŋ_p_*^2^ = 0.125). Further simple effect analysis revealed that, in terms of time, there was no significant difference in SBP between groups at baseline (*p* = 0.149). However, after induced manipulation, the stress group showed significantly higher SBP than the control group (*F*(1,54) = 9.268, *p* = 0.004, *ŋ_p_*^2^ = 0.146), and the stress group also had significantly higher SBP than the control group after the decision-making task (*F*(1,54) = 4.935, *p* = 0.031, *ŋ_p_*^2^ = 0.084). Within the stress group, there were no significant differences in SBP across different time points. In the control group, baseline SBP was marginally higher than that after induced manipulation (*F*(2,53) = 2.943, *p* = 0.061, *ŋ_p_*^2^ = 0.1), but there were no other significant differences.

A two-factor repeated measures ANOVA with a 2 × 3 design was conducted on diastolic blood pressure (DBP) while incorporating the total STAI score as a covariate (Figure 5d). The results indicated a marginally significant main effect of time (*F*(2,53) = 3.091, *p* = 0.054, *ŋ_p_*^2^ = 0.104), with baseline marginally higher than that after the decision-making task (*p* = 0.059) and significantly higher after the induction manipulation compared to after the decision-making task (*p* = 0.031). Additionally, a marginally significant main effect of condition was observed (*F*(1,54) = 3.437, *p* = 0.069, *ŋ_p_*^2^ = 0.06), with the stress group exhibiting marginally higher DBP compared to the control group. There was a marginally significant interaction effect between condition and time (*F*(2,53) = 2.896, *p* = 0.064, *ŋ_p_*^2^ = 0.099). Further simple effect analysis revealed that, in terms of time, there was no significant difference between groups at baseline, but the stress group had significantly higher DBP than the control group after the induction manipulation (*F*(1,54) = 4.544, *p* = 0.038, *ŋ_p_*^2^ = 0.078) and after the decision-making task (*F*(1,55) = 4.121, *p* = 0.047, *ŋ_p_*^2^ = 0.071). Within the stress group, no significant differences were found across time points. However, within the control group, baseline DBP was significantly higher than after the decision-making task (*p* = 0.017), with no other significant differences noted.

#### 3.2.4. Subjective Stress Scores

A two-factor repeated measures ANOVA with a 2 × 3 design was conducted on stress scores while incorporating the total STAI score as a covariate (Figure 5e). The results indicated a significant main effect of time (*F*(2,53) = 6.553, *p* = 0.003, *ŋ_p_*^2^ = 0.198). Specifically, baseline stress scores were significantly lower than those after the induction manipulation (*p* = 0.023) and those after the decision-making task (*p* = 0.001). Additionally, stress scores after the induction manipulation were also significantly lower than those after the decision-making task (*p* = 0.019). The main effect of condition was not significant (*F*(1,54) = 1.655, *p* = 0.204, *ŋ_p_*^2^ = 0.03).

A significant interaction effect was observed (*F*(2,53) = 7.695, *p* = 0.001, *ŋ_p_*^2^ = 0.225). Further simple effect analysis revealed that there were no significant differences between groups at baseline (*p* = 0.408). However, after the induction manipulation, the stress group had significantly higher stress scores than the control group (*F*(1,54) = 6.08, *p* = 0.017, *ŋ_p_*^2^ = 0.17). No significant differences were found between groups after the decision-making task (*p* = 0.176). When analyzing the stress group, a significant main effect of time was observed (*F*(2,53) = 10.759, *p* < 0.001, *ŋ_p_*^2^ = 0.289). Baseline stress scores were significantly lower than those after the induction manipulation (*p* < 0.001) and those after the decision-making task (*p* = 0.001), with no significant difference between the latter two time points. For the control group, a significant main effect of time was also found (*F*(2,53) = 3.382, *p* = 0.041, *ŋ_p_*^2^ = 0.113). However, baseline stress scores were not significantly different from those after the induction manipulation or those after the decision-making task. Notably, stress scores after the induction manipulation were significantly lower than those after the decision-making task (*p* = 0.047).

Taking into account both the psychological and physiological results, it can be concluded that the combination of the social evaluation cold pressor task (SECPT) and the Sing-a-Song Stress Test (SSST) is effective in inducing acute stress, with significant responses observed in both the sympathetic adrenomedullary system and the HPA-axis, but the performance in terms of heart rate is suboptimal.

### 3.3. The Impact of Acute Stress on Ambiguity and Risky Decision-Making

#### 3.3.1. The Results of Farming on Mars Task

A repeated measures ANOVA with a 2 (condition: stress vs. control) × 5 (20-trial blocks) design was conducted to analyze the proportion of selections of the increasing option in the Farming on Mars task, while incorporating the total STAI score as a covariate (Figure 6). The main effect of condition approached significance (*F*(1,54) = 3.385, *p* = 0.071, *ŋ_p_*^2^ = 0.059), with the stress group (0.552 ± 0.04) showing a marginally lower proportion of selections of the increasing option than the control group (0.656 ± 0.039). The main effect of block was significant (*F*(4,51) = 17.109, *p* < 0.001, *ŋ_p_*^2^ = 0.573), with significant differences observed between Block 1 and Block 2 (*p* < 0.001) and between Block 3 and Block 4 (*p* = 0.004). Marginally significant differences were found between Block 2 and Block 3 (*p* = 0.069) and between Block 4 and Block 5 (*p* = 0.069). The interaction effect was not significant.

As a supplementary analysis, independent-samples t-tests were conducted for each block to compare the specific differences between the two conditions across different blocks. In Block 1, the stress group’s proportion of selections of the increasing option (0.411 ± 0.192) was marginally lower than that of the control group (0.503 ± 0.21) (*t*(55) = −1.706, *p* = 0.094, Cohen’s d = −0.45). In Block 4, the stress group’s selection rate (0.602 ± 0.299) was significantly lower than that of the control group (0.748 ± 0.236) (*t*(55) = −2.053, *p* = 0.045, Cohen’s d = −0.55). No significant differences were observed between the two conditions in the other blocks. Additionally, one-way repeated measures ANOVAs were conducted for each block under the two conditions to assess differences in learning between participants under different conditions. In the stress group, significant differences were found between Block 1 and Block 2 (*p* = 0.006) and between Block 4 and Block 5 (*p* = 0.023). In the control group, significant differences were found between Block 1 and Block 2 (*p* = 0.015) and between Block 3 and Block 4 (*p* = 0.002). There were no significant differences observed between the other blocks.

#### 3.3.2. The Results of Cambridge Gambling Task

An independent-samples t-test was conducted to compare the variables under the two conditions. Regarding the risk-adjustment index, the stress group exhibited significantly lower scores (5.66 ± 4.4) than the control group (8.9 ± 6.4) (*t*(55) = −2.22, *p* = 0.031, Cohen’s d = −0.6). However, no significant differences were observed between the groups in terms of quality of decisions, decision time, rational adventure index, risk-taking index, and impulsivity index (Table 1).

#### 3.3.3. The Results of Iowa Gambling Task

A 2 (condition) × 5 (block) repeated measures ANOVA was conducted on the net scores, while incorporating the total STAI score as a covariate (Figure 7). The main effect of block was significant (*F*(4,51) = 9.161, *p* < 0.001, *ŋ_p_*^2^ = 0.418), with a significant difference only between block 1 and block 2 (*p* = 0.003), and no significant differences among the other blocks. The net score, which reflects the difference between the sums of selections from decks 3 and 4 and decks 1 and 2, indicates a higher proportion of favorable decks 3 and 4 being chosen as it increases. Results revealed a significant main effect of block, suggesting that over time, participants in both conditions increasingly selected decks 3 and 4 and decreasingly selected decks 1 and 2, facilitated by learning experience.

Neither the main effect of condition nor the interaction was significant. As a supplementary analysis, independent-samples t-tests were conducted for each of the five blocks sequentially, and no significant differences were found between the stress group and the non-stress group within each block (*p* > 0.1). Additionally, a one-way repeated measures ANOVA was conducted on the net scores of the five blocks under the two conditions. In the stress group, significant differences were found between Block 1 and Block 2 (*p* = 0.008). In the control group, no significant differences were found among the blocks.

### 3.4. Analysis of the Impact of Anxiety on Decision-Making Under Uncertainty

#### 3.4.1. Correlation Analysis (r) Between State Anxiety, Trait Anxiety, and Various Decision-Making Variables

Using Spearman’s correlation analysis, we obtained the correlation between state anxiety, trait anxiety, and various decision-making variables (Table 2). State anxiety was found to have a significant positive correlation only with the net score under ambiguity decision-making conditions (*r* = 0.286, *p* = 0.031). In contrast, trait anxiety exhibited a significant positive correlation solely with the proportion of selections of the increasing option (*r* = 0.289, *p* = 0.029). No significant correlations were observed between the remaining variables and either type of anxiety. Preliminary conclusions suggest that an increase in state anxiety and trait anxiety scores is accompanied by an enhancement in ambiguity decision-making performance.

#### 3.4.2. Analysis of the Moderating Effect of Anxiety

To explore the specific effects of anxiety and stress on decision-making under uncertainty, we employed moderating effect analysis, following the testing procedures proposed by Wen et al. [55]. In this analysis, trait anxiety/state anxiety served as the moderating variable (Z), while the condition (stress versus control) was the independent variable (X). The dependent variables (Y) encompassed various indicators pertaining to ambiguity decision-making and risky decision-making. Hierarchical regression analyses were conducted sequentially. The independent variable was treated as a dummy variable, and the moderating variable was centralized. An interaction term was then created by multiplying the independent variable and the moderating variable. Additionally, age and state anxiety variables were controlled for in the analysis.

Among the results obtained after altering different dependent variables, one positive finding emerged. Specifically, when the dependent variable was the proportion of selections of the increasing option, the results indicated that in Model 3, the interaction term between the condition (X) and trait anxiety (Z) variable exhibited significance (*β* = −0.448, *t* = −2.499, *p* = 0.016 < 0.05) (Table 3). This suggests the presence of a moderating effect, implying that trait anxiety (Z) plays a moderating role between the influence of the condition (X) and the proportion of selections of the increasing option (Y), with statistically significant differences observed at different levels of trait anxiety.

To investigate how trait anxiety moderates the influence of different conditions on the proportion of selections of the increasing option, further simple slope tests were conducted (Figure 8). The trait anxiety scores were divided into high- or low-level groups based on the mean score as well as one standard deviation above and below the mean. According to the slope plot, as trait anxiety scores increase, the effect of the stress condition on the proportion of selections of the increasing option intensifies. At high levels of trait anxiety, the magnitude of the influence (slope) of the independent variable condition on the dependent variable (the proportion of selections of the increasing option) is negative, and the selection rate is significantly higher in the stress condition compared to the control condition (simple slope = −0.202, *t* = −2.293, *p* = 0.026). In contrast, in the low-level group, the slope is positive, but no significant difference is found between different conditions (simple slope = 0.112, *t* = 1.282, *p* = 0.206).

The results indicate that trait anxiety, as a moderating variable, significantly influences the proportion of selections of the increasing option under both stress and control conditions. Specifically, under stress conditions, as trait anxiety scores increase, individuals are more inclined to choose the increasing options that represent long-term rewards. Furthermore, compared to individuals with high trait anxiety, the influence of stress and control conditions on the proportion of selections of the increasing option is weaker among those with low trait anxiety scores.

In addition to this finding, no significant moderating effects of state anxiety and trait anxiety on other decision-making variables have been observed so far.

## 4. General Discussion

In this study, we employed the SECPT in combination with the SSST paradigm to induce acute stress in the laboratory, aiming to investigate the impact of acute stress on decision-making ability under uncertainty and its specific direction. Subsequently, we analyzed the role of anxiety in this process through correlation and the moderating analysis. The research results suggest that acute stress impairs decision-making ability under uncertainty, primarily manifesting as neglect of long-term rewards and reduced risk-adjustment capabilities. Furthermore, higher trait anxiety scores may predict better decision-making performance under ambiguity during stress.

First, following the administration of the SECPT and SSST paradigms, corresponding changes were observed in salivary cortisol levels, heart rate, blood pressure, and subjective stress scores, indicating the successful induction of acute stress. This study provides a viable laboratory example for inducing acute stress. The SECPT, as an effective paradigm for inducing acute stress, has been widely used and proven effective in eliciting a cortisol response, yet there is insufficient evidence of its effectiveness on the sympathetic adrenomedullary system. On the other hand, the SSST has been shown to significantly increase heart rate and galvanic skin response (GSR) indicators, with effects equivalent to those observed within 10 min after the Trier Social Stress Test (TSST) stimulation [49]. However, the SSST also lacks evidence of its impact on the HPA-axis. In our previously unpublished pilot experiment, we found that the SECPT alone had unsatisfactory physiological and behavioral effects on participants in experimental settings. Therefore, this study combined both paradigms to induce acute stress through both the sympathetic nervous system and the HPA-axis, and the results indeed confirm this approach’s effectiveness. However, this acute stress-induction method still has certain limitations. First, the induction of heart rate is not good enough, which may be related to the baseline level of individuals, and needs to be further explored. Additionally, the salivary cortisol was collected at insufficient time points, which only proved the successful induction of stress but did not establish a complete trend of change. Additionally, the analysis would have been more reliable if other indicators such as heart rate variability, skin conductance, and salivary amylase had been included.

To investigate the impact of acute stress on decision-making under uncertainty, we employed the first 40 trials of the Farming on Mars task and the Iowa Gambling Task to analyze ambiguity decision-making, and the last 60 trials of the Cambridge Gambling Task and the Iowa Gambling Task to analyze risky decision-making. In the ambiguity decision-making tasks, participants were not informed beforehand of any potential rewards or reward structures, requiring them to maximize benefits in situations where both probabilities and outcomes were unknown. This part assessed participants’ abilities to engage in inductive reasoning to learn implicit rules and calculate uncertainty, as well as their choices between short-term and long-term rewards. In the risky decision-making tasks, participants were aware of the possible outcomes of the options and the probabilities of risk occurrence, upon which they made decisions by weighing gains and losses. This part involved participants’ ability to adjust strategies according to different risk scenarios and apply learned probabilities and subjective utilities to optimize decision-making.

In the Farming on Mars task, both groups showed an increasing preference for the increasing option over time, with this experiential learning occurring primarily in the early and middle stages of the task. Supplementary analysis revealed that the overall performance of the control group steadily increased, while the stress group exhibited a more pronounced learning effect primarily at the initial stage of the task. Furthermore, the stress group had a lower proportion of selections of the increasing option compared to the control group, indicating that acute stress impaired ambiguity decision-making abilities. Stress slowed down the participants’ acquisition of patterns and led them to prioritize short-term benefits over long-term rewards. During the ambiguity decision-making phase of the IGT, participants showed sensitivity to losses at the initial stage of the task, but there was no significant difference compared to the control group. Consistent with previous research, acute stress caused individuals to prefer smaller, immediate rewards over larger, delayed rewards and enhanced the salience of immediate rewards [28]. However, this finding contradicts some results, which suggested that acute stress promotes the maximization of long-term rewards [26,34].

In the Cambridge Gambling Task, we found that the most significant and sole result was that the stress group scored significantly lower on the risk-adjustment index compared to the control group. Risk-adjustment refers to an individual’s choice of different bet amounts based on changes in reward and punishment probabilities. A lower risk-adjustment ability indicates a tendency towards more risky and impulsive behavior. No significant differences were observed in other variables, suggesting that participants had comparable speeds of information reception and processing, and their probability assessment and response abilities were not significantly affected by acute stress. Therefore, it suggests that participants in the stress group were able to perceive changes in probability information within the task and possessed similar cognitive response abilities. However, they were unable to effectively adjust their betting ratios in response to risk probabilities and lacked the ability to finely tune their risk-taking behavior. Previous studies have proposed that stress can lead individuals to choose more “high-risk, high-reward” options, increasing risk-taking during decision-making [1,24]. Combining these findings with the results of this study, it suggests that acute stress increases risk-taking and leads to a preference for immediate rewards in risky decision-making.

The results found that acute stress reduced individuals’ focus on long-term rewards, impaired their ability to learn patterns, and made them more impulsive and risk-taking. Unable to regulate their risky behaviors, individuals tended to prioritize immediate rewards.

The neurobiological mechanisms underlying the impact of acute stress on decision-making are primarily due to the activation of the sympathetic adrenomedullary system and the HPA-axis, which may increase dopamine release in the prefrontal cortex (PFC) and striatum. This, in turn, may enhance reward sensitivity, thereby influencing decision-making by altering cortical-dependent cognitive processing and reward sensitivity [27,56]. The “STARS” theory proposes that stress enhances individuals’ sensitivity to rewards, improves their ability to learn from positive outcomes, and simultaneously weakens their aversion to losses [27]. In this study, under stress, individuals were slower to extract useful information from feedback on chosen options, while their sensitivity to immediate rewards increased and their sensitivity to long-term rewards decreased. This was partially reflected in the ambiguity decision-making phase of the Iowa Gambling Task. Furthermore, in the Cambridge Gambling Task, individuals under stress became more impulsive and exhibited greater risk-taking, suggesting a reduction in the consideration of potential risks and rewards, an increase in the focus on potential rewards, and a decrease in the aversion to losses. The findings of this study provide evidence for the “STARS” theory. Unlike previous research, this study found that stress primarily enhances sensitivity to short-term rewards rather than long-term rewards. Additionally, since the Iowa Gambling Task in this study did not show significant effects of acute stress, and the Farming on Mars task only involved gain scenarios without loss scenarios, this study could only measure participants’ focus on long-term and short-term gains under stress, and could not get the accurate outcomes of reward and loss feedback in ambiguous situations.

The Farming on Mars task uniquely reflects how one’s past actions influence future rewards, marking a significant distinction from other uncertainty decision-making tasks. Exploring the impact of stress on this type of decision-making holds profound theoretical and practical significance. Previous studies have investigated the influence of stress on such decisions, but the results remain controversial, and they have not jointly examined the impact of stress on ambiguity decision-making in conjunction with risky decision-making. Furthermore, individual differences underlying this impact have not been incorporated into these studies. Therefore, we conducted additional analyses to investigate the role of anxiety as a contributing factor.

In the results of the correlation analysis, it was found that for all participants, an increase in state anxiety and trait anxiety scores was accompanied by an improvement in ambiguity decision-making ability. However, no other significant correlations were observed in the results related to other variables. Based on the results of the moderation effect analysis, it can be suggested that trait anxiety significantly influences the tendency to choose long-term rewards under both stress and control conditions. Specifically, under stress conditions, high levels of trait anxiety prompt the maximization of long-term rewards, while low levels of trait anxiety exhibit lower sensitivity to long-term rewards. This difference is less pronounced under control conditions. This suggests that individuals with high levels of trait anxiety may perform better in ambiguity decision-making under acute stress. Overall, we found that trait anxiety has a more significant impact on ambiguity decision-making under stress, with highly trait-anxious individuals showing a focus on long-term rewards.

This seems to contradict the common perception that anxious and tense emotions can affect the efficiency and accuracy of decision-making, as high trait anxiety often signifies excessive worry and fear, which is not considered positive. However, this study presents an encouraging result for high trait anxiety: in ambiguity decision-making tasks under stress, individuals with high trait anxiety perform better. We delve into the potential reasons underlying our obtained results by examining various factors and considering the context of our study. To provide plausible explanations for the observed outcomes, we present three explanations. First, from a physiological mechanism perspective, the neural mechanism of ambiguity decision-making, mainly involving brain regions such as the amygdala, insula, and PFC, which are also related to the expression and control of fear and anxiety. Studies have observed decreased PFC involvement and altered insula responses in individuals with high trait anxiety, changes that lead to more risk aversion and ambiguity aversion in stressed individuals. Second, in terms of information processing, highly trait-anxious individuals are more attentive to potential risks and negative outcomes, tending to prioritize negative information and thus become more conservative. When faced with more volatile short-term options, they may be more inclined to choose stable but relatively less lucrative long-term options. Finally, in terms of emotional orientation, highly trait-anxious individuals are often more sensitive and intensely affected by negative outcomes, so their avoidance tendency is not only to avoid failure but more so to avoid the emotional stimulus associated with failure. Highly trait-anxious individuals have a stronger ability to learn from rewards but are affected by situations involving losses due to the inclusion of more emotional stimuli. Previous research has shown that highly trait-anxious individuals are easily distracted by task-irrelevant emotional stimuli, reducing their efficiency in complex tasks [35]. However, whether this effect can be deemed as “better” depends on the current situation and specific tasks; sometimes, being overly conservative can be detrimental to certain decisions. Therefore, on the whole, high trait anxiety under stress is not necessarily a bad thing. It can prompt individuals to better consider losses, enhance judgment, and maximize long-term benefits when completing tasks.

This study also has certain limitations:

First, the samples in this study were all male university students to control for the confounding effects of the menstrual cycle. However, existing research has confirmed that estrogen is closely associated with the HPA-axis and stress-related manifestations [29,30,31]. Therefore, conducting research on gender is necessary and future studies should strive to include a more comprehensive and diverse sample population, which would be more beneficial for exploring the relationship between stress and decision-making.

Second, due to the lack of pre-registration and insufficient sample size, which may undermine the robustness and generalizability of the research findings, it is recommended to expand the sample size in future studies in order to bolster the credibility and ensure more reliable and valid conclusions.

Third, the stress situations in our study were conducted in a laboratory setting, which differs significantly from real-life stress. Future studies should design stress scenarios that are closer to reality and more targeted towards different stressors.

Fourth, this study employed an inter-subject design to eliminate significant learning effects within tasks. However, this also led to the existence of individual differences. To mitigate this, we conducted a series of measurements and balancing of participants’ traits (such as personality, anxiety) before the task to weaken such differences. Future studies could also adopt other research paradigms and use an intra-subject design for exploration.

Last, this study investigated decision-making performance under stress from a behavioral perspective. Future research could incorporate other neurophysiological techniques to delve deeper into the underlying mechanisms.

## 5. Conclusions

Most decisions in life are made under uncertainty and acute stress. In this study, we found that acute stress impairs individuals’ ability to learn and make accurate judgments in uncertain decision-making scenarios. Specifically, this manifests as a focus on immediate, smaller rewards and neglect of long-term, larger rewards, accompanied by increased impulsivity and decreased risk management ability. This has a certain enlightening significance for us: In many decision-making environments shrouded in uncertainty, and in situations involving different gains and losses, we need to pay more attention to the immediate and potential consequences of our actions and better control our risk-taking behaviors. Furthermore, higher trait anxiety scores can predict better performance in ambiguity decision-making under stress. This may also provide some practical reference. When a team frequently needs to make decisions under acute stress, perhaps they should have a companion who typically exhibits higher anxiety in daily life nearby. Their vigilance, sensitivity, and conservatism may enhance decision-making in specific situations.

## Figures and Tables

**Figure 1 behavsci-14-01186-f001:**
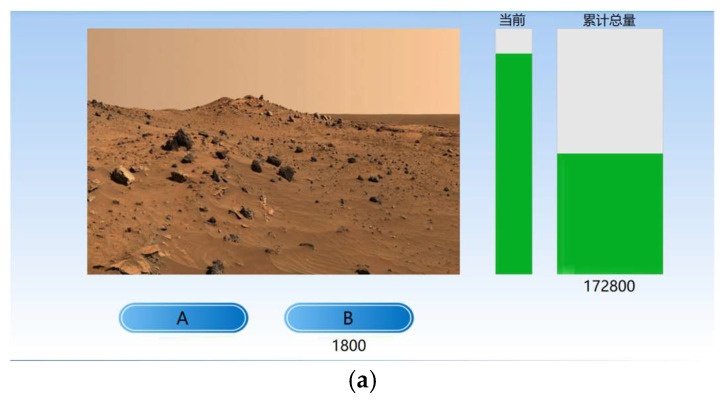
(**a**) Actual interface diagram of the Farming on Mars task. (“当前” means the current option value. “累计总量” means the cumulative total). (**b**) Graph of the task reward score function.

**Figure 2 behavsci-14-01186-f002:**
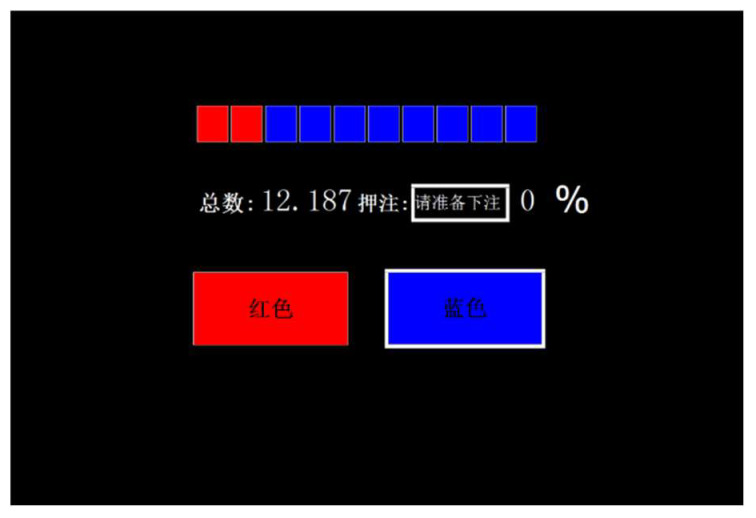
Actual interface diagram of the Cambridge Gambling Task. (From left to right: “总数”—Total, “押注”—Bet, “请准备押注” as “Please prepare to bet”, “红色”—red, “蓝色”—blue) The red and blue squares at the bottom are marked with their corresponding colors).

**Figure 3 behavsci-14-01186-f003:**
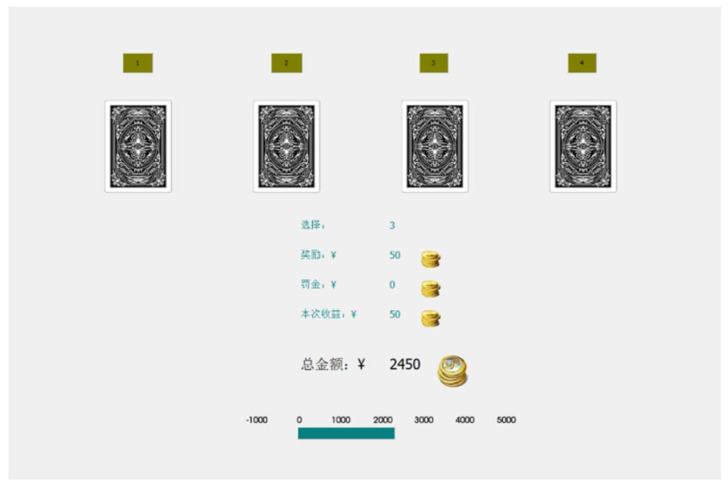
Actual interface diagram of the Iowa Gambling Task. (Green text: “选择”—Selection, “奖励”—Reward Amount, “罚金”—Penalty Amount, and “本次收益”—Current Earnings. Black text: “总金额” represents the total amount).

**Figure 4 behavsci-14-01186-f004:**
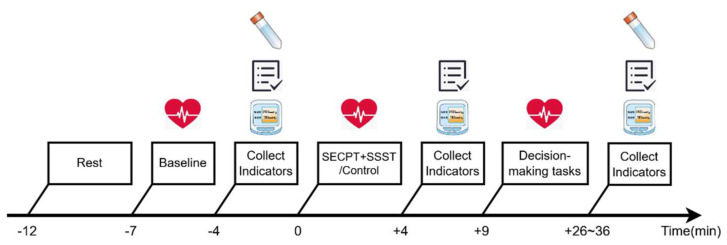
Experimental procedure (this figure graphically illustrates the time periods for heart rate measurement as well as the time points for cortisol assessment, blood pressure measurement, and subjective reporting).

**Figure 5 behavsci-14-01186-f005:**
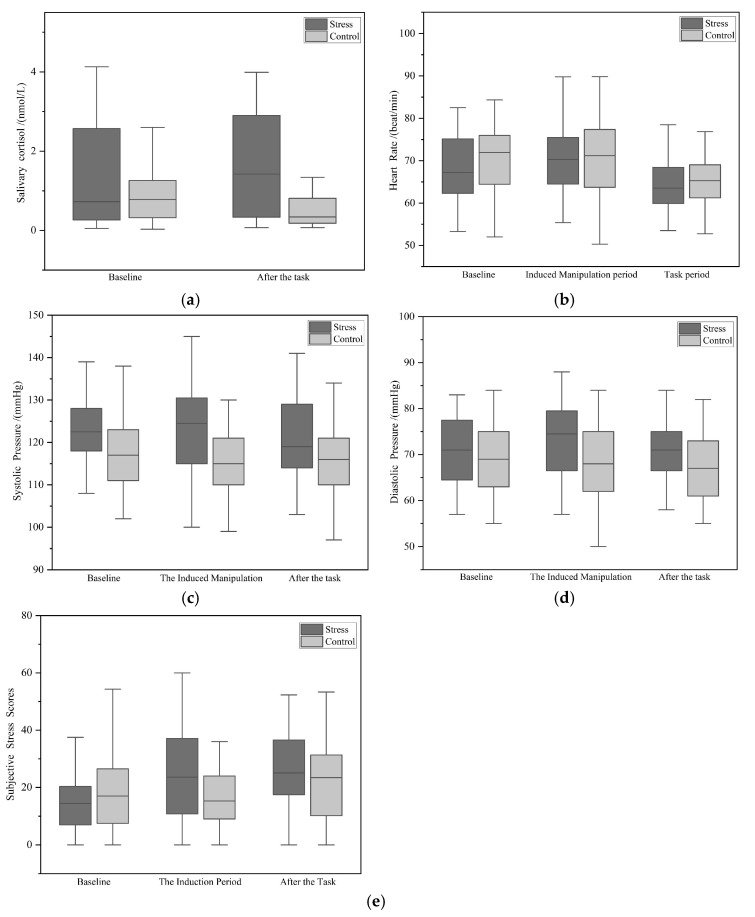
Line graph depicting the results of various indicators of acute stress. (**a**) Salivary cortisol concentration. (**b**) Heart rate. (**c**) Systolic blood pressure. (**d**) Diastolic blood pressure. (**e**) Subjective stress scores.

**Figure 6 behavsci-14-01186-f006:**
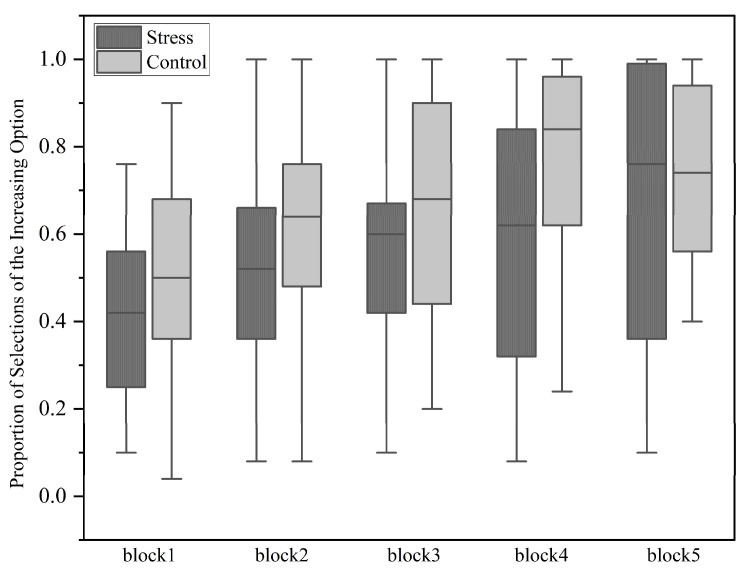
Results of the proportion of selections of the increasing option across five blocks under stress and control conditions.

**Figure 7 behavsci-14-01186-f007:**
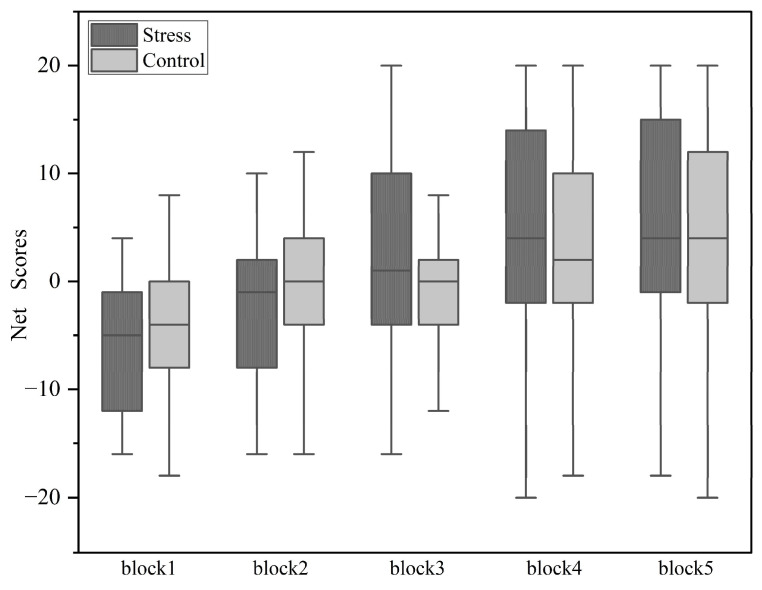
Results of net scores across five blocks under stress and control conditions.

**Figure 8 behavsci-14-01186-f008:**
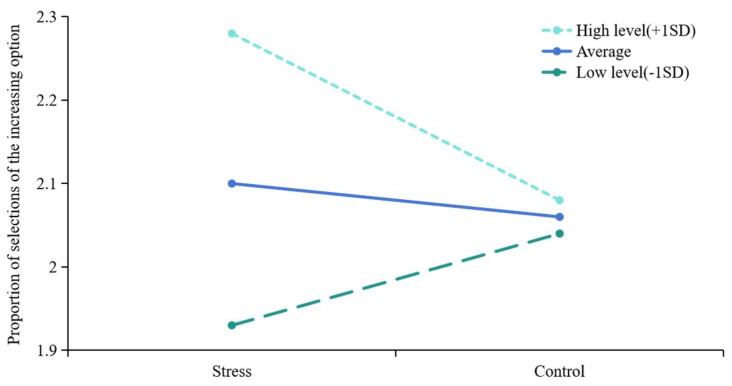
Simple slope plot.

**Table 1 behavsci-14-01186-t001:** Results of independent-samples *t*-tests for various variables in the Cambridge Gambling Task under stress and control conditions.

	Stress	Control	*t*	*p*
Quality of Decisions	29.36 ± 3.87	29.83 ± 4.25	−0.436	0.664
Decision Time	1.82 ± 0.82	1.67 ± 0.61	0.738	0.464
Rational Adventure Index	0.48 ± 0.16	0.46 ± 0.16	0.382	0.704
Adventure Index	0.45 ± 0.15	0.43 ± 0.15	0.432	0.667
Impulse Index	0.12 ± 0.13	0.12 ± 0.12	0.043	0.966
Risk-adjustment	5.66 ± 4.4	8.9 ± 6.4	−2.22	0.031 *

* Significant at the *p* = 0.05 level.

**Table 2 behavsci-14-01186-t002:** Results of Spearman’s correlation analysis between state anxiety, trait anxiety, and various decision-making variables.

	State Anxiety	Trait Anxiety
Proportion of Selections of the Increasing Option	−0.21	0.289 *
Net Score in Ambiguity decision-making	0.286 *	−0.043
Quality of Decisions	−0.08	−0.203
Decision Time	0.161	0.185
Rational Adventure Index	0.216	−0.042
Risk-taking Index	0.208	−0.079
Impulsivity Index	−0.166	−0.047
Risk-adjustment	0.153	0.141
Net Score in Risky decision-making	−0.118	−0.067

* Significant at the *p* = 0.05 level.

**Table 3 behavsci-14-01186-t003:** Results of moderating effect analysis.

	Model 1		Model 2		Model 3	
	*β*	*t*	*β*	*t*	*β*	*t*
Constant	-	3.413 **	-	3.982 **	-	4.285 **
State Anxiety	−0.177	−1.331	−0.301	−2.208 *	−0.352	−2.676 **
Age	−0.233	−1.765	−0.213	−1.69	−0.187	−1.551
**Independent variable**						
Condition	−0.032	−0.243	−0.088	−0.684	−0.09	−0.731
**Moderating variable**						
Trait Anxiety			0.346	2.498 **	0.697	3.616 **
Condition × Trait Anxiety					−0.448	−2.499 *
*R* ^2^	0.081		0.18		0.269	
*F*	*F*(3,53) = 1.567, *p* = 0.208		*F*(4,52) = 2.852, *p* = 0.033		*F*(5,51) = 3.761, *p* = 0.006	
△*R*^2^	0.081		0.098		0.089	
△*F*	*F*(3,53) = 1.567, *p* = 0.208		*F*(1,52) = 6.242, *p* = 0.016		*F*(1,51) = 6.246, *p* = 0.016	

* Significant at the *p* = 0.05 level. ** Significant at the *p* = 0.01 level.

## Data Availability

The data presented in this study are available on request from the corresponding author. The reason why the data cannot be published is that the experiment has not been fully completed yet, and the data are temporarily not suitable for public disclosure.

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
