# Peer review of "Trait Anxiety Leads to “Better” Performance? A Study on Acute Stress and Uncertain Decision-Making"

_behavsci, 2024, doi:10.3390/bs14121186_

Round 1

Reviewer 1 Report

Comments and Suggestions for Authors

Dear Authors,

thank you very much for your manuscript. I  generally like the idea to study decisions under uncertainty and to relate these to further factors such as - in your case - stress and anxiety. That said, however, I have several reservations about the paper, related to referencing, research design and interpretation of results (please see below for details). I will recommend a revise and resubmit, albeit expressing concerns about eventual publication.

The comments below are organized by section. Minor comments are mentioned at the end.

Abstract:

Background - that sounds very unspecific and not very convincing.

Results - Sentence 1, I have no idea what this is saying.

Introduction:  

My overall impression of the introduction was that it is rather unorganized  in that paragraphs do not properly build on each other and the motivation for the study is lacking clear focus. Moreover, and most critically, I find the number of references (compared to the claims made) dramatically insufficient (examples of lines where references are missing for the claims made: 43, 44, 46, 60, 79, 83).

Sentence starting line 30 is saying what?

Paragraph starting line 58 is particularly unorganized. The last sentence of it appears to be a good topic sentence for the paragraph.

Reference 11 refers to mice, if I got that correctly.

Line 97: I strongly advise against statements about the own research being meaningful. Either it is - then this should be clear from the motivation. Or it is not - then emphasizing  it won’t make it better.

Where do the hypotheses at the end come from?

I would suggest a brief preview of results and possibly a roadmap for the paper.

Materials and Methods

I would have expected a section heading to be rather “Methods and Procedures”.

I find the number of subjects very small given the number of tasks and variables measured, and I find the focus on men problematic. While I see why you focus on men, that at least ought to be mentioned in the introduction and abstract already as it severely limits the scope of your results.

Line 113: Questionnaires asking for what?

Are there references for the stress inducing tasks?

Line 134: ”indifferent experimenter” - I think I see what you mean but I doubt that there is anything like an indifferent experimenter.

Related Scales: references missing!

Please refer to the figures in the text. Otherwise, it is not immediate why they are there. Please test the colors for black/white printing.

Line 175ff: References?

Line 188: References?

Line 227: Refernces?

Were the tasks all presented in the same order? 

Results

Line 423: may be I am missing something, but what does block 1-4 refer to?

Line 426-429: I find that confusing. What is the relation to the arguments in the introduction?

Line 436 (and later): I would suggest to be very careful about the interpretation of the results (not least in view of the small number of subjects). With that I refer to expressions like ‘can be inferred’ (here), ‘is’ (line 595)… I agree that these are possible interpretations but would prefer “suggest” or similar as you mostly use anyway.

Line 443: I would suggest a new paragraph. The sentence would be a nice summary.

Line 606: you differentiate between gains and losses. Why here? Why not mentioned before? What are relevant references? That gain/loss framing matters a lot is well known.

In general: In view of a lack of preregistration and insufficient derivation of hypotheses, I think your result that “trait anxiety lead to better performance” might be due to chance. I am not saying it is, but please consider that possibility.

Discussion

Line 695ff.: I did not understand that part.

Limitations: Please mention the small sample, lacking pre-registration and many variables (including the possibility for subject fatigue).

Conclusion

That does not say much, does it?

Author Response

Dear Reviewer,

Thank you very much for your thorough and insightful comments on my manuscript.  I deeply appreciate the time and effort you have invested in providing such detailed feedback.Your suggestions have been extremely helpful in enhancing the clarity, accuracy, and overall quality of my manuscript. The following is my response to your comments.

Comments 1 : Abstract: Background - that sounds very unspecific and not very convincing. Results - Sentence 1, I have no idea what this is saying.

Response 1 : I’ve revised the abstract to make it more specific in its description. Further details can be found in the background and results sections of the abstract.

Introduction:  

Comments 2 : My overall impression of the introduction was that it is rather unorganized  in that paragraphs do not properly build on each other and the motivation for the study is lacking clear focus.

Response 2 : In order to make the introduction more organized, I reorganized the paragraphs and added some transition sentences and concluding paragraphs. I hope this will make the reading experience smoother for both you and the readers.

Comments 3 : Sentence starting line 30 is saying what?

Response 3 : The non-specificity of stressful responses refers to the uniform stress syndrome exhibited by different individuals upon exposure to any stressor, such as the activation of the HPA axis. However, given the complexity of stressful responses, not all stressors elicit entirely identical nonspecific reactions. Consequently, the specificity of stressful responses denotes the potential for varied changes in the stress reaction, resulting from differences in stressors or individual variations under the same stressor. I have modified the sentence for better understanding. Please see line 37 for details.

Comments 4 :Moreover, and most critically, I find the number of references (compared to the claims made) dramatically insufficient (examples of lines where references are missing for the claims made: 43, 44, 46, 60, 79, 83).

Response 4 : Thank you for your meticulous and patient pointing out. I have added the corresponding references one by one.

Comments 5 :Paragraph starting line 58 is particularly unorganized. The last sentence of it appears to be a good topic sentence for the paragraph.

Response 5 : I have reorganized the language and content of this section. This section primarily provides a sequential review of the impact of acute stress on ambiguity decision-making and risky decision-making. For further details, please refer to Section 1.3.

Comments 6 :Reference 11 refers to mice, if I got that correctly.

Response 6 : For the sake of rigor, I have replaced the references. Please see reference [17].

Comments 7 :Line 97: I strongly advise against statements about the own research being meaningful. Either it is - then this should be clear from the motivation. Or it is not - then emphasizing  it won’t make it better.

Response 7 : I have revised the wording.

Comments 8 :Where do the hypotheses at the end come from?

Response 8 : The hypotheses were derived based on the review of the existing literature, theoretical frameworks, and empirical evidence related to our research topic. I have modified the hypotheses section to make it more detailed.

Materials and Methods

Comments 9 :I would have expected a section heading to be rather “Methods and Procedures”.

Response 9 : I have made modifications accordingly.

Comments 10 :I find the number of subjects very small given the number of tasks and variables measured.

Response 10 :Thank you for your point. We consider that although there are relatively more measurement tasks and variables, all tasks were completed within-subjects, with only the conditions being between-subjects. Regarding the issue of sample size arising from between-subjects grouping, we have referenced previous similar studies for guidance, noting that the sample sizes ranged from 40 to 113, as mentioned in the Participants section. Finally, we acknowledge that the sample size poses limitations on the generalization of our research findings, and we have accordingly added this point in the Limitations section of our paper.

Comments 11 :And I find the focus on men problematic. While I see why you focus on men, that at least ought to be mentioned in the introduction and abstract already as it severely limits the scope of your results.

Response 11 :I fully understand your concerns, and I have provided detailed descriptions of gender considerations in both the section 1.3 and the limitations section, along with the reasons for our choices.

Comments 12 :Line 113: Questionnaires asking for what?

Response 12 : The questionnaire assessed a series of scales, including but not limited to the Chinese Big Five Personality Inventory-Brief Version (CBF-PI-B) (which were not the primary outcomes of this study). Specifically, the results from the CBF-PI-B, along with those from the State-Trait Anxiety Inventory (STAI), were utilized to analyze whether the grouping was balanced and whether the two groups of participants came from the same population. The CBF-PI-B is mentioned in the sections related to Procedures and Balance Analysis. For better understanding, I have added descriptive details in line 188.

Comments 13 :Are there references for the stress inducing tasks?

Response 13 : The stress-induction paradigm employed in our study is a combination of the Socially Evaluation Cold Pressure Test(SECPT) with the Sing-a-Song Stress Test(SSST). After I have revised the order of the references, the citation for SECPT is [46], and the citation for SSST is [47] However, there is currently no literature that combines the two paradigms.

Comments 14 :Line 134: ”indifferent experimenter” - I think I see what you mean but I doubt that there is anything like an indifferent experimenter.

Response 14 :I fully understand your concerns about the term "indifferent experimenter." In our study, under stressful conditions and based on previous research, we established a condition where a neutral experimenter observes from the sidelines. It is important to clarify that the "indifferent experimenter" does not imply that the experimenter is unfeeling or uncaring. Rather, during the experiment, the experimenter strives to maintain an objective and neutral stance, avoiding providing additional emotional support or interference to the participants, thereby ensuring the objectivity and accuracy of the experimental results. To avoid misunderstanding, I have revised the manuscript, replacing "indifferent" with "neutral."

Comments 15 :Related Scales; Line 175ff; Line 188; Line 227: references missing

Response 15 : I have added references at the corresponding lines. For details, please refer to lines 216, 256, 269 and 308.

Comments 16 :Please refer to the figures in the text. Otherwise, it is not immediate why they are there.

Response 16 : Thank you for your meticulous attention and pointing out the issues. I have added them one by one.

Comments 17 :Were the tasks all presented in the same order? 

Response 17 : The three decision-making tasks were presented in a Latin square order to control for any potential effects due to the order of the tasks. For instance, Participant 1 completed the tasks in the order of 123, Participant 2 in 231, Participant 3 in 312, and so on. There were multiple task orders, and each participant was randomly assigned one. This is mentioned on line 317.

Results

Comments 18 :Line 423: may be I am missing something, but what does block 1-4 refer to?

Response 18 :Blocks 1 and 4 refer to the trials ranging from the 1st to the 50th and from the 150th to the 200th, respectively, within the Farming on Mars task, which comprises a total of 250 trials. To facilitate analysis and observe trends in variation, we have divided these trials into five blocks. For the convenience of readers in understanding, I have incorporated additional descriptions into section 2.3.1 (on line 237). Additionally, due to modifications made in the discussion section, I have removed this part.

Comments 19 :Line 426-429: I find that confusing. What is the relation to the arguments in the introduction?

Response 19 : The primary focus of research on decision-making under uncertainty is how individuals make choices in uncertain situations to maximize their rewards. I apologize for not describing it clearly enough, which may have caused confusion; it was due to a lack of emphasis on our part. Therefore, I have included descriptions of short-term and long-term benefits at the beginning of the introduction and in section 1.3.

Comments 20 :Line 436 (and later): I would suggest to be very careful about the interpretation of the results (not least in view of the small number of subjects). With that I refer to expressions like ‘can be inferred’ (here), ‘is’ (line 595)… I agree that these are possible interpretations but would prefer “suggest” or similar as you mostly use anyway.

Response 20 : We express our sincere respect for your rigorousness. I will exercise greater caution in interpreting the results, and I have revised the sections in the text that involve the interpretation of the results.

Comments 21 :Line 606: you differentiate between gains and losses. Why here? Why not mentioned before? What are relevant references? That gain/loss framing matters a lot is well known.

Response 21 : I have added relevant content concerning losses and gains, as detailed in Section 1.3.

Comments 22 :In general: In view of a lack of preregistration and insufficient derivation of hypotheses, I think your result that “trait anxiety lead to better performance” might be due to chance. I am not saying it is, but please consider that possibility.

Response 22 : I have added issues such as the lack of pre-registration and insufficient sample size in our limitations section. I fully understand that the results you mentioned could be attributable to chance, and we acknowledge the shortcomings of our research. But I approve of the rigor and meticulousness of our experimental procedures, and I maintain my confidence in the authenticity and accuracy of our current results. Even if we step back and assume that the results were indeed due to chance, I believe that every chance occurrence contains an element of necessity, and every necessity manifests through numerous chance events. There is no such thing as pure chance or pure necessity. Isn't research about uncovering genuine necessities amidst various chance occurrences?

Discussion

Comments 23 :Line 695ff.: I did not understand that part.

Response 23 : This part contains my explanations for the potential causes of the results. I have added some introductory remarks preceding this part, as detailed on line 686-689.

Comments 24 :Limitations: Please mention the small sample, lacking pre-registration and many variables (including the possibility for subject fatigue).

Response 24 :I have provided supplementary explanations for these limitations.

Conclusion

Comments 25 :That does not say much, does it?

Response 25 : I have expanded the conclusion section.

Reviewer 2 Report

Comments and Suggestions for Authors

Trait Anxiety Leads to "Better" Performance? A study on acute stress and uncertain decision-making is a study using Latin square design for the order of three tasks measuring decision making under uncertainty. The farming on mars is an optimization problem, the Cambridge gambling task measures risk taking / aversion and the Iowa gambling task has both ambiguous and risk elements. The authors also measured 3x cortisol (saliva), blood pressure and heart rate. The study has some potential if the analysis is done correctly, and there would have been only one group to have sufficient power.

There are major issues in the analysis and result section

heart rate: independent t -test is clearly wrong. You test within a single group the three pairs (baseline vs induction, baseline vs post-task, induction vs post-task) - that would require paired t-test as these are the same participants. You will also need to correct for multiple testing

you have to report degrees of freedom and for follow-up tests the effect sizes

figure 5 & 6 lacks error bars or confidence intervals, consider using boxplots as they show the distribution, median and quartiles

line 370: one-sample t-test but then you report a comparison between the two groups (... was marginally lower than that of the control group ...), so it must be independent t-test (not paired t-test). One-sample t-test is if you compare it to a reference value, i.e. .5

line 397ff: please report not just that there was a difference between blocks, but whether the choice for deck C&D vs deck A&B went up or down.

3.3.4 has to come later. The introduction is set up to look at anxiety, so far a standard stress on decision-making study is reported. All previous sections should use the STAI scores as co-variate

if you want to keep it separate, then you have to correct for multiple testing (table 2) by using e,g. FDR or Holm correction

line 470ff: you treat the STAI scores (state / trait) as predictors and the scores from the tasks as outcome, but you need to also control for condition (group), i.e. stress or no stress condition. You have not established that the groups score similar on train and state anxiety (only that they score similar on Big 5). Also your previous analysis has shown that stress condition has an effect on the task scores. Hence, you now want to find out whether the difference is larger / smaller after controlling for state / trait anxiety. That is how you set it up in the introduction. Your regression analysis is not answering that.

for example, from table 5 I can see that train anxiety affects negatively the quality of decision in the control group, but positively in the stress group. But I don't know whether the quality of decision is affected more by stress vs control or by trait anxiety. But that is - according to your introduction - what you want to study. Strictly speaking, for your research question (anxiety effect on "uncertain decision-making" under stress) you would not need a control condition. Your design should have been (more statistical power!) one condition (the stress only) and measuring trait and state anxiety. Furthermore, of the physiological measures maybe only heart rate variability (indicator of how stressful it is) would be needed, but both genders. You may well find that anxiety's effect on decision-making under stress is different for men and women. Not least when it comes to risk. 

minor:

line 104 / 113 / 114: make clear the 20.25 is years

line 141ff: are the reliability scores McDonald's omega? Or Cronbach's alpha? Please report

make also clear whether you used average score or sum score

line 160: you state the DV is proportions - di you transform the variable before using in the analysis?

line 175: how did you calculate the rational adventure index?

line 207: zona fasciculata of the adrenal cortex

line 223 and other places in the manuscript: since you use 3 tasks, it is less confusing if you refer to it as "after the decision-making tasks", and not as "after the task" - that can be confused with after the intervention or sing-a-song task. Thanks

line 249: the data was analyzed

line 252: an independent t-test ... please correct, this is incorrect, see above

line 262: better to say that they came from the same population (instead of being homogenous)

table 6 is not needed if you write it out in table 7 and the equations. But see other comments to rectify the entire analysis

Author Response

Dear Reviewer,

I am extremely grateful for your comprehensive and accurate comments on my manuscript. Your expertise and keen eye for detail have been incredibly helpful in refining my work. Thank you once again for the time and effort you have invested in improving my research. I look forward to continuing to learn and grow as a researcher under the guidance and support of such a respected reviewer as yourself.

Comments 1 :heart rate: independent t -test is clearly wrong. You test within a single group the three pairs (baseline vs induction, baseline vs post-task, induction vs post-task) - that would require paired t-test as these are the same participants. You will also need to correct for multiple testing

Response 1 : Due to our oversight, the error occurred in the analysis. We sincerely appreciate your bringing it to our attention, and we have made the necessary corrections. Given the complexity involved in correcting for multiple paired-sample t-tests, and considering that our re-analysis revealed a marginally significant interaction effect, we have re-examined the heart rate data using a simple effect analysis. This approach not only facilitates a thorough comparison of differences between different levels but also significantly reduces the risk of committing a Type I error. For further details, please refer to section 3.2.2.

Comments 2 :you have to report degrees of freedom and for follow-up tests the effect sizes

Response 2 : I have supplemented the missing degrees of freedom and effect sizes at the corresponding locations.

Comments 3 :figure 5 & 6 lacks error bars or confidence intervals, consider using boxplots as they show the distribution, median and quartiles

Response 3 : I have re-created the figures as boxplots, which indeed convey much more information and look better now. Thank you very much for your suggestion. For details, please refer to Figures 5 to 7.

Comments 4 :line 370: one-sample t-test but then you report a comparison between the two groups (... was marginally lower than that of the control group ...), so it must be independent t-test (not paired t-test). One-sample t-test is if you compare it to a reference value, i.e. .5

Response 4:I am deeply sorry for the mistake made during the preparation of our article. We actually employed the independent-samples t-test as our analytical method, but due to our oversight, we incorrectly stated it as a one-sample t-test. I have now corrected it to "independent-samples t-test." Once again, I express my admiration for your expertise and proficiency in statistics.

Comments 5 :line 397ff: please report not just that there was a difference between blocks, but whether the choice for deck C&D vs deck A&B went up or down.

Response 5:I have supplemented the results in the section 3.3.3, reporting the trends in the selection of decks 3&4 and 1&2.

Comments 6 :3.3.4 has to come later. The introduction is set up to look at anxiety, so far a standard stress on decision-making study is reported. All previous sections should use the STAI scores as co-variate

Response 6: I have removed this part and merged it into a single comprehensive discussion section. Additionally, I have incorporated the STAI scores as covariates in all previous results analyses. For further details, please refer to the corresponding results.

Comments 7 :if you want to keep it separate, then you have to correct for multiple testing (table 2) by using e,g. FDR or Holm correction

Response 7: I have revised it to conduct a combined analysis.

Comments 8 :line 470ff: you treat the STAI scores (state / trait) as predictors and the scores from the tasks as outcome, but you need to also control for condition (group), i.e. stress or no stress condition. You have not established that the groups score similar on train and state anxiety (only that they score similar on Big 5). Also your previous analysis has shown that stress condition has an effect on the task scores. Hence, you now want to find out whether the difference is larger / smaller after controlling for state / trait anxiety. That is how you set it up in the introduction. Your regression analysis is not answering that.

for example, from table 5 I can see that train anxiety affects negatively the quality of decision in the control group, but positively in the stress group. But I don't know whether the quality of decision is affected more by stress vs control or by trait anxiety. But that is - according to your introduction - what you want to study. Strictly speaking, for your research question (anxiety effect on "uncertain decision-making" under stress) you would not need a control condition. Your design should have been (more statistical power!) one condition (the stress only) and measuring trait and state anxiety.

Response 8: We are deeply grateful for your insightful comments and for granting us the opportunity to make revisions. The points you raised are extremely pertinent, and your explanations are thorough and detailed. We have sincerely adopted your suggestions and altered our analytical approach, re-analyzing the results accordingly. We have employed analysis of moderating effects, which allows us to consider both the impact of stress and the role of anxiety within it. Unfortunately, we obtained fewer results than previously, but this, in turn, enhances the authenticity and reliability of our findings. We are unsure whether our analysis is entirely correct, and we would welcome any criticism or corrections you may have. We sincerely appreciate your valuable feedback.

Comments 9 :Furthermore, of the physiological measures maybe only heart rate variability (indicator of how stressful it is) would be needed, but both genders. You may well find that anxiety's effect on decision-making under stress is different for men and women. Not least when it comes to risk. 

Response 9: Regarding the measurement of heart rate variability (HRV), regrettably, the device we utilized currently lacks an algorithm for HRV analysis, rendering an analysis of HRV unfeasible. Therefore, we have incorporated additional indicators such as salivary cortisol and blood pressure for comprehensive analysis. In terms of gender considerations, I have included related content on this topic in both the introduction and the limitations section. In the future, we will seriously consider your suggestions and conduct research on individual performance across different genders and scenarios, with a particular emphasis on the HRV indicator.

Comments 10 :line 104 / 113 / 114: make clear the 20.25 is years

Response 10: I have added annotations indicating the ages of the participants in section 2.1.

Comments 11 :line 141ff: are the reliability scores McDonald's omega? Or Cronbach's alpha? Please report

Response 11: I have included the Cronbach's alpha coefficient for the reliability of the scale. For details, please refer to line 217.

Comments 12 :make also clear whether you used average score or sum score

Response 12:The subsequent analysis focuses on the total scores, and I have clarified this in the text. For further details, please refer to line 221.

Comments 13 :line 160: you state the DV is proportions - di you transform the variable before using in the analysis?

Response 13:The results show the frequencies of choosing options A or B. For the purpose of facilitating analysis, we adopted the dependent variable as the proportion of the number of times option A was chosen to the total number of choices. This approach is consistent with previous research in the field. To enhance reader understanding, I have included additional explanations regarding the dependent variable. For further details, please refer to line 240.

Comments 14 :lline 175: how did you calculate the rational adventure index?

Response 14:Thank you for your meticulous observation. The Rational Adventure Index refers to the average betting percentage for choosing the majority color (excluding 5:5 scenarios). I have made the necessary correction. For details, please refer to line 259.

Comments 15 :line 207: zona fasciculata of the adrenal cortex

Response 15:I have revised it accordingly.

Comments 16 :line 223 and other places in the manuscript: since you use 3 tasks, it is less confusing if you refer to it as "after the decision-making tasks", and not as "after the task" - that can be confused with after the intervention or sing-a-song task. Thanks

Response 16: Thank you for your meticulous attention. I have sequentially changed "after the task" to "after the decision-making task" within the manuscript. However, due to word limit constraints, the captions in the figures still retain the term "after the task."

Comments 17 :line 249: the data was analyzed

Response 17:I have revised it accordingly.

Comments 18 :line 252: an independent t-test ... please correct, this is incorrect, see above

Response 18:I have changed the analysis method and revised the terminology accordingly.

Comments 19 :line 262: better to say that they came from the same population (instead of being homogenous)

Response 19: I have revised it accordingly.

Comments 20 :table 6 is not needed if you write it out in table 7 and the equations. But see other comments to rectify the entire analysis

Response 20: Due to the change in methodology, I have removed this table.

Round 2

Reviewer 1 Report

Comments and Suggestions for Authors

Dear Authors,

thank you very much for addressing the comments and sending an updated version. My impression is that the paper has improved considerably. However, I have some further comments which I believe need to be addressed. These are not many, though, and if the quality of the revision remains constant, I believe that the next version should be acceptable for publication.

Please find detailed comments below (I put a * before comments which I believe should be addressed but which are not critical); generally I found the formatting (little space between text and figures / tables) unfortunate:

*Abstract: “This paper aims to explore…” … sounds like really a lot with eventually few data. Perhaps “..contribute to the exploration…” or similar?

Paragraph starting line 62: I think that requires more referencing.

Paragraph starting Line 95:  “from us”??  “Real of ecomics”…are gains and losses not relevant everywhere? The end of the paragraph was confusing for me. Transition to the next was awkward.

Paragraph starting line 115, End:

The part with gender is not the primary focus is unfortunate. I would suggest something like “As geneder effects are unclear and, moreover, difficult to control for women, we focus on on male participants in the sequel.” Or similar.

*Paragraph starting line 157:

I would suggest to delete everything up to “…, this study aims..” and put a “Contributing to the discussion,…” before.

Line 216: Reference 48 covers all scales? 

Line 588f: If you refer to old research, can you reference that?

Line 592-603: For one thing, I would suggest to start a new paragraph. Then I think the grammar is wrong “has the limitations”. Finally, why does the limitations part come there? I found that part confusing.

*Line 623: I’d suggest starting a new paragraph.

*Line 639: I’d suggest starting a new paragraph.

Author Response

Dear Reviewer,

Thank you once again for your review and further suggestions for revision. Your meticulousness and rigorousness are truly commendable and deeply touch us. Based on your recommendations, I have made the necessary adjustments to the inappropriate parts. Below are my responses.

Comments 1: *Abstract: “This paper aims to explore…” … sounds like really a lot with eventually few data. Perhaps “..contribute to the exploration…” or similar?

Response 1: I have modified it accordingly.

Comments 2: Paragraph starting line 62: I think that requires more referencing.

Response 2: I have added two additional references at the appropriate locations to provide citations for the concepts.

Comments 3: Paragraph starting Line 95:  “from us”??  “Real of ecomics”…are gains and losses not relevant everywhere? The end of the paragraph was confusing for me. Transition to the next was awkward.

Response 3: My original intention was to emphasize the primary focus of this study. However, due to misinterpretations, we have decided to omit "from us." And I have changed "in the realm of economics" to "from an economic standpoint."

Furthermore, regarding transitional sentences, at the conclusion of this paragraph, we summarize the chosen paradigm and the reasons for selecting it based on the content described above. Additionally, I have revised the opening sentence of the next paragraph to echo this one, thereby introducing another key variable.

Paragraph starting line 115, End:

Comments 4: The part with gender is not the primary focus is unfortunate. I would suggest something like “As gender effects are unclear and, moreover, difficult to control for women, we focus on male participants in the sequel.” Or similar.

Response 4: I have revised the last sentence of the paragraph, making it more precise and concise indeed. For details, please refer to line 129-131.

*Paragraph starting line 157:

Comments 5: I would suggest to delete everything up to “…, this study aims..” and put a “Contributing to the discussion,…” before.

Response 5: I have made the modifications according to your suggestions. For details, please refer to line 155.

Comments 6: Line 216: Reference 48 covers all scales?

Response 6: The State-Trait Anxiety Inventory (STAI) is a scale that comprises two subscales, with scores calculated separately for each. Consequently, they are introduced within the same reference.

Comments 7: Line 588f: If you refer to old research, can you reference that?

Response 7: Thank you for your constructive suggestions. We fully understand the importance of providing clear and traceable sources for all information presented in academic writing. However, in this particular case, due to the fact that the pilot study has not undergone peer review and publication, we are unable to provide a traditional citation. To ensure transparency and reproducibility, we have included in the manuscript a corresponding explanation of the chosen methods, clarifying that they are based on preliminary results from our unpublished pilot experiment.

Comments 8: Line 592-603: For one thing, I would suggest to start a new paragraph. Then I think the grammar is wrong “has the limitations”. Finally, why does the limitations part come there? I found that part confusing.

Response 8: I have made modifications to this section. Firstly, I have changed “has the limitations” to “has certain limitations”. Subsequently, I would like to clarify the reasons for presenting these limitations. These limitations are specifically raised in relation to the stress-induction paradigm we have chosen, and represent further analysis and discussion of this induction method. To avoid ambiguity and enhance the logical flow of the reading, I have revised “our study” to “this acute stress-induction method” to explicitly state that the limitations are associated with this particular method. And merging these points into a single paragraph serves to present them as a cohesive whole. For details, please refer to line 589.

Comments 9:*Line 623: I’d suggest starting a new paragraph.; *Line 639: I’d suggest starting a new paragraph.

Response 9: I have carefully considered your suggestion. Your proposal is well-taken, and indeed, this paragraph was somewhat overly long. To enhance logical flow and readability, I have redistributed the paragraphs. A new paragraph has been initiated at line 624, with lines 610-624 constituting a summary discussion of ambiguity decision-making, lines 625-640 presenting a summary discussion of risky decision-making, and lines 641-644 offering a summary of the overall results concerning decision-making under uncertainty and stress.

Thank you very much for your insights and understanding.

We wish you all the best!

Reviewer 2 Report

Comments and Suggestions for Authors

The revision was done thoroughly and greatly improved the paper - not least by redoing some of the analysis.

Well done

Author Response

Dear Reviewer,

We express our gratitude for your recognition, and extend our best wishes for your continued success and well-being.

Best Regards,

All authors

Round 3

Reviewer 1 Report

Comments and Suggestions for Authors

Dear Authors,

thank you very much for the detailed response to my comments and for integrating the suggested changes. I will recommend acceptance of your paper.

Good luck with your research.

PS: typo in Line 116 "gender as" supposedly should be "gender is".